# Milmed Yeast Alters the LPS-Induced M1 Microglia Cells to Form M2 Anti-Inflammatory Phenotype

**DOI:** 10.3390/biomedicines10123116

**Published:** 2022-12-02

**Authors:** Federica Armeli, Beatrice Mengoni, Elisa Maggi, Cristina Mazzoni, Adele Preziosi, Patrizia Mancini, Rita Businaro, Thomas Lenz, Trevor Archer

**Affiliations:** 1Department of Medico-Surgical Sciences and Biotechnologies, Sapienza University of Rome, Corso della Repubblica 79, 04100 Latina, Italy; 2Department of Biology and Biotechnologies “C. Darwin”, Sapienza University of Rome, Piazzale Aldo Moro 5, 00185 Roma, Italy; 3Department of Experimental Medicine, Sapienza University of Rome, Viale Regina Elena, 324, 00161 Roma, Italy; 4Milmed Unico AB, 11139 Stockholm, Sweden

**Keywords:** yeast, millimeter wavelength, microbiota, microglia, M1/M2 polarization, neuroinflammation, anti-inflammatory

## Abstract

Microglial cells polarized towards a proinflammatory phenotype are considered the main cellular players of neuroinflammation, underlying several neurodegenerative diseases. Many studies have suggested that imbalance of the gut microbial composition is associated with an increase in the pro-inflammatory cytokines and oxidative stress that underlie chronic neuroinflammatory diseases, and perturbations to the gut microbiota were detected in neurodegenerative conditions such as Parkinson’s disease and Alzheimer’s disease. The importance of gut-brain axis has been uncovered and the relevance of an appropriate microbiota balance has been highlighted. Probiotic treatment, rebalancing the gut microbioma, may reduce inflammation. We show that Milmed yeast, obtained from *S. cerevisiae* after exposure to electromagnetic millimeter wavelengths, induces a reversal of LPS-M1 polarized microglia towards an anti-inflammatory phenotype, as demonstrated morphologically by the recovery of resting phenotype by microglia, by the decrease in the mRNAs of IL-1β, IL-6, TNF-α and in the expression of iNOS. Moreover, Milmed stimulated the secretion of IL-10 and the expression of Arginase-1, cell markers of M2 anti-inflammatory polarized cells. The present findings data suggest that Milmed may be considered to be a probiotic with diversified anti-inflammatory activity, capable of directing the polarization of microglial cells.

## 1. Introduction

Evidence from animal studies indicates a role of probiotics in regulating brain homeostasis, counteracting inflammation underlying neurodegenerative diseases and depression [1,2,3,4]. Neuroinflammation is now hypothesized to be the key mechanism of Alzheimer’s disease (AD), induced by the polarization of microglial cells towards a proinflammatory phenotype. Microglial cells play a central role in maintaining brain homeostasis and are involved in resolving inflammation from trauma or infectious microorganisms by means of phagocytosis and/or anti-inflammatory mediators [5]. It has been recently shown that in the experimental models of AD treated with probiotics it was possible to observe a reduction in inflammatory processes as well as an increase in the level of antioxidant enzymes, and a decrease in β-amyloid deposition as well as tau hyperphosphorylation [6]. As a matter of fact, many studies have suggested that imbalances of the gut microbial composition are associated with an increase in the pro-inflammatory cytokines and oxidative stress that underlie chronic neuroinflammatory diseases [7]. The importance of the gut-brain axis has been recently uncovered and the relevance of a correct microbiota balance has been highlighted [8]. An imbalance in the components of the microbiota can damage the intestinal barrier, affecting in a bidirectional way the CNS; indeed, microbiota alterations were shown to be directly associated with neuropsychiatric disorders, promoting the development of depression and dementia [4]. Many studies have suggested that imbalances of the gut microbial composition are associated with an increase in the pro-inflammatory cytokines and oxidative stress that underlie chronic neuroinflammatory diseases [7]. In addition, perturbations to the gut microbiota were detected in neurodegenerative conditions such as Parkinson’s disease (PD) [9] and Alzheimer’s disease (AD) [10], a complex community of microorganisms influenced not only by host genetics, but also by diet and the environment [11]. Moreover, it has recently been hypothesized that microorganisms may contribute to the development of Alzheimer’s disease, either by producing amyloid-like molecules, enhancing the production of endogenous Aβ, or by increasing systemic inflammation that, targeting glial cells, contributes to the neuronal damage observed during Alzheimer’s disease [12].

In this connection, much interest was focused on probiotics, the beneficial living bacteria and yeast, that may rebalance the bacterial-fungal gut microbiome and reduce inflammation [13,14,15]. Several studies have reported the beneficial effect of *Saccharomyces* yeast supplementation, decreasing inflammation and oxidative stress. In 2021 Durmaz et al. [16] reported that oral supplementation of probiotic *S. boulardii* before supraceliac aortic ischemia-reperfusion in rats alleviates lung injury by reducing oxidative stress, intestinal cellular damage, and modulation of inflammatory processes. Of note, *S. boulardii* supplemented to mice to whom gut dysbiosis was induced by oral Ampicillin Na, was able to reduce oxidative stress and inflammatory cytokines and chemokines resulting in hippocampal neuron protection and eventually reversing gut dysbiosis associated with cognitive decline in mice [17]. Moreover, three strains of *S. cerevisiae* isolated from milk kefir showed to be proper candidates as probiotic yeast strains for the development of novel functional foods [18] since they were able to colonize and adhere to epithelial intestine-derived cells, thanks to their resistance and survival ability in the gastrointestinal physiological conditions.

So far, the yeast *Saccharomyces boulardii* has been studied extensively for its beneficial effects upon health and thereby for its probiotic functions, while information on other probiotic yeast strains remains limited. A strain of *Saccharomyces cerevisiae* isolated from “nuruk” (a traditional Korean fermentation initiator) has recently been proposed as an effective probiotic, one which is able to influence the innate and adaptive immune system, reducing in particular the levels of mRNAs for IL-1β and IFNγ [15]. *Saccharomyces boulardii* was supplemented to APP/PS1 mice, an AD model transgenic mice characterized by neuroinflammation dependent on microglia activation and the TLRs pathway, that after treatment recovered from dysbiosis and improved their cognitive impairment [19].

The purpose of present study was to assess the ability of Milmed to counteract the microglia polarization towards the M1 pro-inflammatory phenotype induced by LPS and to analyze Milmed’s propensity to counteract inflammation. Our previous results showed that Milmed *Saccharomyces cerevisiae* yeast [Milmed], treated with electromagnetic waves in the Extreme High Frequency (EHF) range of 30–300 GHz, does not produce cytotoxic effects, either upon microglia cells or upon cells of neuronal origin [2]. Moreover, when added to LPS-stimulated microglial cells, it promoted a remodeling of the cells that acquired the branched spindle-like morphology characteristic of quiescent unstimulated cells, losing some phylopodia. If added to human neuroblastoma cells, Milmed yeast promoted the expression of BDNF and NGF mRNAs [2]. Furthermore, patients presenting allergic problems aged from 30 to 71 years-of-age received Milmed supplementation for 4 to 13 weeks. The median treatment time was 9 weeks. After the test period, each patient rated the perceived improvement of allergies on a scale of 1 to 10 and the perceived improvement in general health on a scale of 1 to 10. There were both self-assessed allergy level significant improvements (8.38 ± 0.62) and self-assessed general health level improvements (7.38 ± 0.82) in comparison with the control group. Milmed supplementation to allergy patients alleviated allergy symptoms related to the number of weeks the treated Milmed was ingested [20]. In the present study, the polarization of microglial cells was analyzed, measuring the cytokine and chemokine mRNAs expression, as well as the expression of M1 and M2 markers such as iNOS and Arginase-1 by real-time PCR and immunofluorescence assays.

## 2. Materials and Methods

### 2.1. Yeast Strains and Growth

The treatment and preparation of *S. cerevisiae* with electromagnetic waves in the Extreme High Frequency (EHF) range of 30–300 GHz produces a treated yeast extract, given the name Milmed [21,22,23,24].

This treatment was developed through the pioneering work of Golant and co-workers [25,26,27] on *S. cerevisiae*, a strain that was obtained originally from the International Research Center ‘‘Beer and Beverage XXI Century’’ (Moscow, Russia), but now from Sweden. The yeast was cultured in wort which was produced through meaded malt extract. The treatment of the yeast in an electromagnetic field of super-high frequencies with electromagnetic waves in the EHF range of 30–300 GHz produced the treated yeast extract [19], after which the yeast was re-cultured at 25–28 °C for 48 h [28]. Un-treated and treated strains from Milmed AB (www.milmed.de, accessed on 1 November 2022) were grown under shaking overnight at 28 °C in YP (1% yeast extract, 2% bactopeptone; DIFCO), supplemented with 2% glucose or 3% glycerol, SD (yeast nitrogen base w/o amino acids 0.67%) glucose 2% or glycerol 3% media. Cells were harvested at exponential phase and the number of cells quantified by spectrophotometry at 600 nm. 1. OD = 13.33 × 10^6^ cells/mL.

For each experiment, 10^7^ cells were collected and washed in PBS. To verify temperature inactivation, yeast cells incubated at 60 °C for 15 min, were centrifuged for 5 min at 3200 rpm to collect cells, plated on YP plus agar 2% for 24 h and examined for the eventual colony formation.

### 2.2. Cell Culture and Treatment

The BV-2 murine microglial cell line, kindly provided by Dr. Mangino, Sapienza University of Rome (Italy), was cultured in Dulbecco’s modified Eagle’s medium (DMEM, Euroclone, Pero, MI, Italy), supplemented with 10% fetal bovine serum (FBS; Sigma-Aldrich, St. Louis, MO, USA) and 1% penicillin-streptomycin (Sigma-Aldrich, St. Louis, MO, USA), at 37 °C in a humidified incubator under 5% CO_2_, until they reached 90% confluence. Cells were seeded in 6 well plates (cell density of 10^6^ cells/well) and incubated at 37 °C with 5 × 10^2^ of treated or untreated yeast grown in different media (YP 2% Glucose, or YP 3% Glycerol, SD 2% glucose) for 45 min, after 45 min BV-2 cells were treated with lipopolysaccharide (LPS strain 0111:B4, Sigma Aldrich, St. Louis, MO, USA) 1 ng/mL and compared to control.

### 2.3. Cell Viability Assays

#### 2.3.1. Trypan Blue Exclusion Assay

Cell viability was determined by Trypan-Blue exclusion assay. Trypan-blue (Euroclone, Pero, MI, Italy) exclusion assay is a simple and rapid method measuring cell viability that determines the number of viable cells and dead cells. It is based on the principle that live cells with an intact membrane are able to exclude the dye whereas dead cells without an intact membrane take up the dye. For the Trypan blue exclusion test, BV-2 were seeded onto 48-well plates at a density of 3 × 10^4^/well. After treatments, cells were detached with 1 × Tripsin-EDTA (Sigma-Aldrich, St. Louis, MO, USA), and 10 μL of cell suspension were mixed with 10 μL of Trypan Blue solution and cell counts were performed using a Burker chamber. Blue stained cells were considered nonviable.

#### 2.3.2. Immunofluorescence Microscopy

Cell cultures were stimulated with LPS, a prototypical microbial antigen in the presence or absence of Milmed in different dilutions. Briefly, BV2 cells grown in chamber-slides were incubated with Milmed grown in YP 2% Glucose, or YP 3% Glycerol, or SD 2% glucose or SD 3% glycerol, in the presence or in the absence of LPS for 24 h. Cells were subsequently fixed in 4% paraformaldehyde in phosphate-buffered-saline (PBS) for 30 min at 25 °C, followed by treatment with 0.1 M glycine in PBS for 20 min at 25 °C and with 0.1% Triton X-100 in PBS for additional 5 min at 25 °C to allow permeabilization. To analyze cytoskeletal actin reorganization, microglial cells were incubated with Phalloidin-Tetramethylrhodamine B isothiocyanate (Phalloidin-TRITC) (Sigma-Aldrich, St. Louis, MO, USA), 1:50 for 45 min at 25 °C. For the detection of M1/M2 polarization markers, cells were incubated with rabbit polyclonal antibodies raised against iNOS (dil. 1:100—D6B65, Cell Signaling Technology, Danvers, MA, USA) or rabbit polyclonal IgG anti-Arg-1 (dil. 1:50—D4E3M, Cell Signaling Technology, Danvers, MA, USA) and subsequently with anti-rabbit Alexa Fluor 488 secondary antibodies. Finally, the cells were marked with DAPI (Sigma-Aldrich, St. Louis, MO, USA), to highlight the nucleus. The fluorescence signal was analyzed using an Axio Observer inverted microscope, equipped with the ApoTome System (Carl Zeiss Inc., Ober Kochen, Germany). Cell area was quantified with ImageJ software 1.48 version.

### 2.4. Real-Time Quantitative PRC Analysis

Total RNA was extracted from the control and treated BV-2 cells using the miRNeasy Micro kit (Qiagen, Hilden, Germany) and quantified using NanoDrop One/OneC (Thermo Fisher Scientific, Waltham, MA, USA). cDNA was generated using the High-Capacity cDNA Reverse Transcription kit (Applied Biosystem, Foster City, CA, USA). Quantitative real-time PCR (qPCR) was performed for each sample in triplicate on an Applied Biosystems 7900HT Fast Real-Time PCR System (Applied Biosystem, Cheshire, UK) through the program SDS2.1.1 (Applied Biosystem, Foster City, CA, USA) using the Power SYBR^®^Green PCR Master Mix (Applied Biosystem, Foster City, CA, USA). The primers for real-time PCR amplification were designed with UCSC GENOME BROWSER (http://genome.cse.ucsc.edu/ (accessed on 1 November 2022); university of California, Santa Cruz, CA, USA) (Table 1). The primer pair sequences were matched by BLASTn to the genome sequence to identify the primer locations with respect to the exons. A comparative threshold cycle (C_T_) method was used to analyze the real-time PCR data, where the amount of target, normalized to the endogenous reference of β-Actin (∆CT) and relative to the list of primer couples generated for qPCR.

### 2.5. Statistical Analysis

Data were expressed as the mean values ± standard deviations (SD) from at least three independent experiments. The statistical significance was determined by one-way ANOVA analysis coupled with a Bonferroni post-test or unpaired *t*-test (GraphPad Prism 5.0 software, GraphPad Software Inc., La Jolla, CA, USA) and defined as * *p* < 0.05, ** *p* < 0.01, and *** *p* < 0.001. Ns is not significant. Parametric statistics, one-way ANOVA and Bonferroni tests. were applied to the results since group size was held constant throughout and normal distributions were obtained.

## 3. Results

In order to assess the anti-inflammatory capacity of Milmed, we examined the expression of these markers after stimulation with LPS in the presence or absence of Milmed in different forms (dried or cultivated in medium with different metabolic characteristics). Accordingly, we studied the metabolic characteristic of Milmed yeasts with the intention of examining for a possible effect of differential metabolic yields. Yeast was grown in rich medium YP (Yeast, Peptone) containing all the metabolites needed for yeast viability and in SD medium (Synthetic Defined, without amino acids) containing only ammonium sulphate as nitrogen base, in which yeast is forced to activate anabolic pathways for amino acids synthesis. Moreover, we supplemented YP with glucose 2%, which allows only fermentation, or glycerol 3%, a respiratory carbon source, to analyze the yeast’s catabolic properties. In YP glycerol 3% the growth was slower compared to YP glucose 2%, as expected for brewer’s yeast. Our previous data showed that any toxicity was observed by Trypan blue staining on BV2 microglia cell cultures incubated with a concentration to 5 × 10^2^ Milmed/well [2].

### 3.1. BV-2 Cellular Areas

Since morphological changes represent a fast event of microglial cell activation, the first topic we examined in this paper was to quantify the effect of LPS in the presence or not of Milmed on cellular phenotypes. In a previous report [2], we have shown that in resting conditions the cells have a classical microglial phenotype, with a central cell body from which different cellular branches start. When the cells are stimulated, they are activated and take on an enlarged, rounded and flat morphology, with retraction of the cellular protrusions. Based on this result, here we attempted to quantize these changes. To this purpose, we evaluated the area of the BV-2 cells grown under the same conditions as in the previous work [2], performing immunofluorescence analysis using phalloidin staining of F-actin to highlight the cellular morphology.

Untreated yeast added to cells did not change their cellular area (Figure 1A), which kept to a size similar to that of control cells; however, after stimulation with LPS there was an increase in cell areas as compared to control.

When LPS-treated BV-2 cells were supplemented with MILMED (Figure 1B), there was a reduction in cell area that reverted to control phenotype, suggesting that yeast promotes the recovery of a resting phenotype.

When BV-2 cells were supplemented with MILMED (Figure 1B) there was a reduction in cell area that reverted LPS treatment, suggesting yeast’s role in recovering a resting phenotype.

### 3.2. iNOS and Arg-1 Expression

When activated, microglia are switched onto either a neurotoxic M1- or a neuroprotective M2-activated phenotype, depending upon the type of stimuli [29]. To evaluate here the M1 or M2 status of the BV-2 cells, we performed both immunofluorescence and RT-qPCR analyses and assessed the level of nitric oxide synthase (iNOS) and Arginase-1 (Arg-1) phenotype, which are markers for the M1 and M2 microglia phenotypes, respectively.

Figure 2 depicts the staining associated with iNOS and ARG-1 in BV2 cells following different treatments. Figure 2A: LPS was able to stimulate the expression of iNOS, M1 marker, that was almost undetectable in untreated cells. The immunofluorescence results show a cytoplasmic dotted signal. It is also possible to observe variations in the cell shape: untreated BV2 are smaller, while after incubation with LPS they acquire an amoeboid form with increased cell body diameters, as expected for activated microglial cells and as demonstrated previously in the quantization of the cellular area (Figure 1). The most effective treatment in contrasting the expression of iNOS is that carried out with Milmed yeast grown in 3% glycogen, which also allows a recovery of the quiescent phenotype. The same result, even if at a lower level, is observed in the treatments with Milmed yeast grown in SD 2% Glu, as shown in the graph that reports the fluorescence intensity/cell. Small fluctuations in values observed after the addition of yeasts grown in different culture conditions confirms that the observed effects can be attributed directly to yeasts and their metabolites.

Arginase-1, a marker of M2 anti-inflammatory cells, was induced by the yeast addition. As shown in Figure 2B, Milmed grown in YP Gly 3% and in SD Glu 2% are very effective by themselves, their activity is increased when cells are stimulated by LPS. On the other hand, untreated yeast does not stimulate the expression of ARG-1.

Therefore, Milmed does not induce any inflammatory effect by itself; on the contrary, it stimulates the synthesis of ARG-1 mRNA, a marker of M2, anti-inflammatory phenotype.

These results were confirmed by analyses of iNOS and ARG-1 mRNAs obtained from BV2-cells stimulated by LPS in the presence or absence of yeast supplied as dried preparation or grown in YPD medium (see Figure 2A,B). Yeast by itself does not induce iNOS mRNA, and instead decreases iNOS mRNA stimulated by LPS. Furthermore, Milmed alone is able to induce the synthesis of ARG-1 mRNA and to partially recover its decrease after LPS addition, skewing microglia towards an anti-inflammatory phenotype (see Figure 3).

### 3.3. Cytokines Expression

We also evaluated the production of proinflammatory cytokines such as IL-1beta, IL-6 and TNF-alpha and anti-inflammatory cytokines, such as IL-10 in the presence of Milmed (in liquid or dried form) and after stimulation with LPS. As shown in Figure 4, Milmed, especially in the dried form, was able to counteract the expression of IL-6 mRNA. Also, the expression of mRNA for TNF alpha was reduced markedly in the presence of Milmed yeast that, by itself, was completely lacking the ability to induce the synthesis of these two proinflammatory cytokines.

Therefore, Milmed counteracts LPS activity since the levels of mRNAs IL-1b, IL-6, TNF-alpha measured by qRT-PCR were reduced in the presence of the yeast. (Figure 4A–C). Finally, Milmed, whether in suspension or in the dried form, promoted the expression of the anti-inflammatory cytokine, Il-10 (see Figure 4D) thereby further confirming the anti-inflammatory actions of the millimeter wavelength treated yeast.

## 4. Discussion

Microbiota is involved in regulating gastrointestinal (GI), immune, nervous system and metabolic homeostasis [30]. When disruptive alterations in the composition of the microbiota, termed dysbiosis, take place, an imbalance of gut microbiota associated with unhealth outcome affects not only the health and integrity of the gastrointestinal tract but also promotes the onset of neuropsychiatric/neurologic diseases [4,31]. Dysbiosis of gut microbiota can induce increased intestinal permeability, immune activation leading to systemic inflammation, which in turn may impair the blood-brain barrier and promote neuroinflammation, neural injury, and ultimately, neurodegeneration [32,33]. Several chronic neurodegenerative diseases affecting the central nervous system have been associated with the polarization of microglia towards a proinflammatory phenotype; these are characterized generally by a change in the shape of the cells and by increased expression of iNOS, as well as an increased synthesis of proinflammatory cytokines [34,35,36,37,38]. The interconnections of gut microbiota perturbation with CNS disorders have been well-documented in recent times, as well as the real and/or putative interactions with the immune system [39,40,41]. Furthermore, the direct impact of gut microbiota on the functions of microglial cells has been shown [39,42]. Activated microglial cells play a central role in the development of many CNS diseases, including depression, AD, and PD [38,43,44,45,46,47]. Their release of cytokines and chemokines target neuronal cells leading to cellular dysfunctionality and eventual cell death [48]. Probiotic agents augment an improvement of depression symptoms and stress-related diseases modulated by stress hormones and cytokines release [49] in conjunction with microbiota influencing the ability of neurons, brain tissue and immune cells to regulate inflammation [50]. Clinical and experimental data have indicated that the probiotics induce positive impacts upon the central nervous system [51,52]. Probiotics were shown to reduce CNS inflammatory processes and activation of microglia, decreasing neuroinflammation, probably through the production of the brain-derived neurotrophic factor (BDNF) and the improved synaptic plasticity, related to an increased concentration of postsynaptic density protein 95 (PsD95) [53] A randomized, double-blind, and controlled clinical trial on AD patients showed that a 12-week probiotic treatment induced a significant improvement in the MMSE score and a significant decrease in plasma malondialdehyde in the probiotic-treated group [54].

Yeast supply to an experimental model of depression prevented depression-like behavior induced by stress and inflammatory challenges in mice [15]. Stress may lead also to neuroimmune dysregulation of inflammation through microbial dysbiosis [7]. Chronic stress and early life stress affect the structure and function of the gut microbiota, which in turn affects immunoregulatory responses and the microbiota– gut-brain axis [55].

The present results imply that the Milmed preparation of *S. cerevisiae,* treated with electromagnetic wavelengths in the Extreme High Frequency—millimeter range [2,22,28,56], present a probiotic agent that drives the innate immune cells’ polarization disposition. According to this scenario, the decrease in cytokine release after LPS inflammatory stimulus and the polarization of microglial cells towards an anti-inflammatory phenotype, as confirmed by the induction of IL-10 and arginase-1, following Milmed addition, leads to the assertion that Milmed supplementation offers a promising treatment for pathologies arising from neuro-inflammation, such as depression, Parkinsonism and AD. Our results agree with data published by Feng et al. [57] on the regulation of macrophage polarization by yeast microcapsules. In that paper, yeast was shown to facilitate macrophagic propulsion towards an M1 or M2 phenotype. It has been demonstrated that beta-1,3-D-glucan, β-D-glucose polysaccharides that are naturally-occurring in the cell walls of fungi and bacteria, on yeast surface is recognized and bound by dectin-1 receptor and complement receptor3 expressed on innate immune cells: after binding yeast cells are phagocytized and their components enter into the immune cells [58].

As outlined by Palepu and Dandekar in their recent paper [39] and by other reports [59] the cross-connection of gut bacteria with the expression of BDNF in the brain, and intestinal microbiota supplementation may produce an antidepressant effect via augmentation of BDNF. In our previous report, we showed that Milmed yeast addition to human SK-N-SH neuroblastoma cells promoted the expression of BDNF and NGF, confirming that Milmed co-treatment may be suitable for several neurologic and neuropsychiatric disorders [2]. Thus, remodeling of the microbiota-gut-brain axis using psychologically active probiotics appears to be a promising therapeutic approach for the amelioration of neuropsychiatric disorders.

Moreover, numerous recent reports have highlighted the possibility of using compounds of vegetable origin, known as nutraceuticals, or particular diets, to treat inflammation and to drive macrophage polarization towards an anti-inflammatory phenotype [60,61,62,63]. The evidence of a microbiota–gut–brain axis suggests that modulation of the gut microbiota may be a tractable strategy for developing novel therapeutics for complex CNS disorders, and that dietary habits play a crucial role in the selection of the bacterial community in the human gastrointestinal tract [60] Plant-derived molecules such as resveratrol, were shown to target macrophage-related inflammation associated with neurodegenerative diseases; they were shown to suppress the LPS-stimulated microglia release of proinflammatory mediators such as NO, TNF-α, iNOS, IL-1β, and IL-6, associated to M1 proinflammatory phenotype, and switch the cells to the expression of Arg1, CD163, and IL-10, markers associated with the M2 anti-inflammatory phenotype [64]. Moreover, resveratrol was shown to counteract the 7-oxo-cholesterol-triggered proinflammatory signaling in macrophages [65]. Several nutritional protocols, including Mediterranean diet and ketogenic diets, were able to counteract neuroinflammation in cellular experimental models as well as in clinical trials [60,61,62,63,66,67,68].

Much interest is now focused on the ability of the microbiota and in particular of some yeast strains to promote health, as shown by recent papers dealing with experimental models and clinical trials concerning chronic inflammatory diseases [4]. In addition, their application acquires particular relevance not only for their survival rate in the gut but also for their antibiotic resistance [69]. The new *Saccharomyces* strains, such as the SC28-7 strain isolated from the traditional Korean nuruk, a starter for brewing alcoholic beverages, show beneficial effects for health parameters. This probiotic yeast supplement for gut health is able to recover immune response and mucosal dysfunction in mice colitis [70]. *Saccharomyces cerevisiae* var. *boulardii* has been used for the treatment of inflammatory gastrointestinal diseases and its anti-inflammatory properties have been linked to the interference with transcription factor nuclear factor κB (NF-κB), considered the main transcription factor of proinflammatory genes, and the modulation of the mitogen-activated protein kinases ERK1/2 and p38 signaling pathways [71]. The influence of *S. boulardii* on TNF-α and IL-6 regulation was also investigated using different colon cell lines, showing that its addition reduces the cytokine levels [72]. 

Our results show that Milmed exerts an anti-inflammatory activity on microglia cells, the innate immunity cells that play a central role in the development of some neurological and neuropsychiatric diseases, as demonstrated by the decrease in the mRNAS of IL-1β, IL-6, TNF-α and expression of iNOS after treatment with LPS. Furthermore, Milmed yeast has the ability to polarize microglial cells in an anti-inflammatory sense as it stimulates the secretion of IL-10 and the expression of Arginase-1, cell markers of M2 anti-inflammatory polarized cells. The present data findings suggest therefore that Milmed may be considered to be a probiotic with diversified anti-inflammatory activity, capable of directing the polarization of microglial cells. These preclinical findings require confirmation by both preclinical and randomized clinical trials to demonstrate its full efficacy among patients. The data collected so far, despite the limitation of the low number of subjects enrolled in the studies, have shown that Milmed provides a beneficial effect in subjects suffering from allergies and IBS-IBD Conditions [20,73].

## 5. Conclusions

A treated yeast, Milmed, has been shown in several studies to exert anti-inflammatory actions against the proinflammatory action of LPS, as indicated above. These present findings indicate that Milmed has induced both (i) an anti-pro-inflammatory effect, i.e., reduction of iNOS that LPS exposure had caused, and (ii) a direct anti-inflammatory action arising from enhanced ARG-1 expression. Further anti-inflammatory effects were induced through the antagonism of the pro-inflammatory effects of the pro-inflammatory cytokines, IL-1β, IL-6 and TFN-α, together with the enhanced expression of the anti-inflammatory cytokine, IL-10.

Limitations: The status of Milmed as a probiotic agent requires characterization with regard to distribution and fate in vivo.

### Future Perspectives

Gut microbiota and their metabolites cooperate to the maintenance of homeostasis by interacting with innate and adaptive immune components; therefore, microbiome-targeted therapies such as probiotics, prebiotics and synbiotics hold great potentials to prevent and treat inflammatory diseases. The anti-inflammatory effects of a treated yeast preparation, Milmed, provide a plausible therapeutic application in the treatment of several areas of infection and/or allergic reactions, as recently indicated [17,40]. In view of the anti-neurodegenerative observations of the treated yeast [21], further research upon the effects of the treated yeast upon growth factors and autophagy ought to be explored.

## Figures and Tables

**Figure 1 biomedicines-10-03116-f001:**
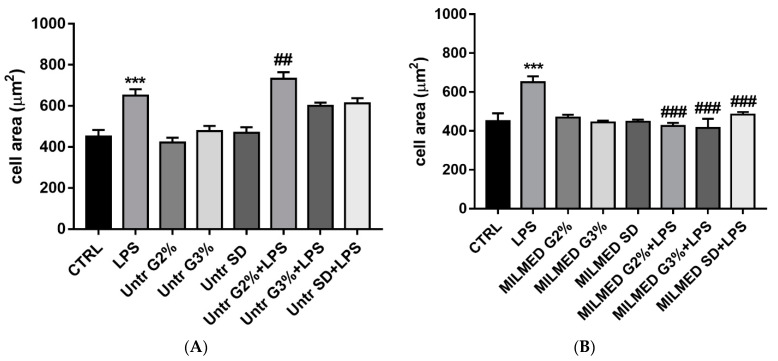
(**A**) BV2 cells area following Lipopolysaccharide (LPS) treatment (1 ng/mL) for 24 h in the presence or absence of untreated yeast grown in different cultural conditions. LPS induced a cell body enlargement which was even more evident in the presence of untreated yeast grown in G2%. (**B**) BV2 cells area following LPS treatment (1 ng/mL) for 24 h in the presence or absence of Milmed: after Milmed addition cells showed a measure of area similar to that of control in Glucose 2%, Glycerol 3% and SD with LPS. Data are expressed as mean ± SD for each group (*n* = 3). Statistical analysis was performed by the one-way analysis of variance (ANOVA) method coupled with the Bonferroni post-test. *** *p* < 0.001; ## *p* < 0.01; ### *p* < 0.001.

**Figure 2 biomedicines-10-03116-f002:**
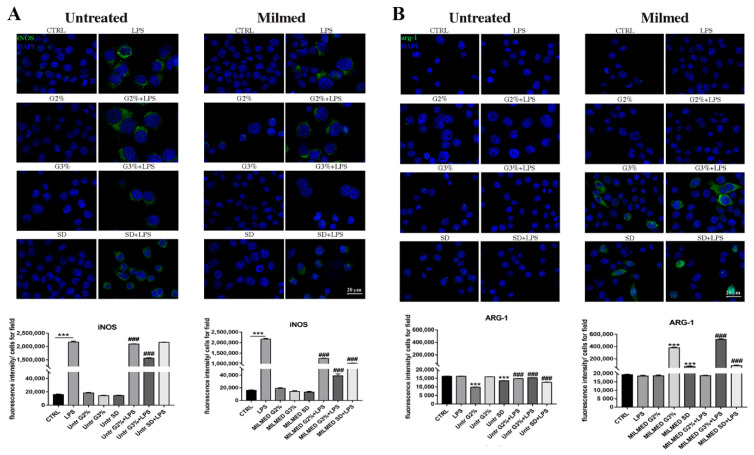
(**A**) Immunofluorescence analysis of Inducible Nitric Oxide Synthase (iNOS) expression in BV-2 cells cultured in the presence of untreated yeast grown for 24 h in different metabolic medium without LPS (on the left, first column) or after addition of Lipopolysaccharide (LPS) 1 ng/mL (second column) and (**B**) Arginase-1 (ARG-1)expression in BV-2 cells cultured in the presence of Milmed grown in different metabolic medium without LPS (on the right, first column) or after addition of LPS 1 ng/mL (second column). Quantification of the median fluorescence intensity was performed by ImageJ software and data were expressed as histograms, normalized to the number of cells for field. 4′,6-diamidino-2-phenylindole (DAPI)was used to counterstain the nuclei. Data are expressed as mean ± SD for each group (*n* = 3). Statistical analysis was performed by the one-way analysis of variance (ANOVA) coupled with the Bonferroni post-test. *** *p* < 0.001; ### *p* < 0.001.

**Figure 3 biomedicines-10-03116-f003:**
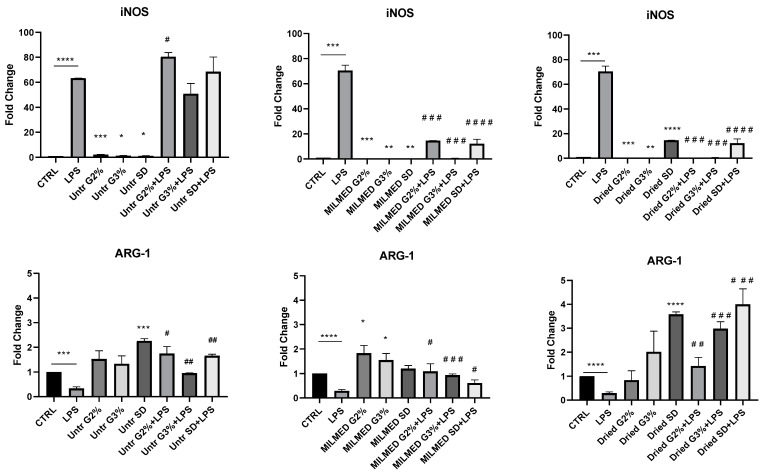
BV2 cells were pretreated with 5 × 10^2^ Milmed/well grown in Yeast Extract–Peptone–Dextrose (YPD) medium or regrown in YPD medium from a dried preparation for 45 min and afterward incubated with Lipopolysaccharide (LPS) 1 ng/mL. Inducible Nitric Oxide Synthase (iNOS) and Arginase-1 (ARG-1) mRNAs expressions were evaluated by qRT-PCR at 4 h and normalized to β-actin. Data are shown as mean ± SD from three independent experiments performed in triplicate. Statistical analysis was evaluated by unpaired Student’s *t* test. Expression profiles were determined using the 2^−ΔΔCT^ method. * *p* < 0.05, ** *p* < 0.01. *** *p* < 0.001, **** *p* < 0.0001; ## *p* < 0.01; ### *p* < 0.001; #### *p* < 0.0001 * VS CTRL, # VS LPS.

**Figure 4 biomedicines-10-03116-f004:**
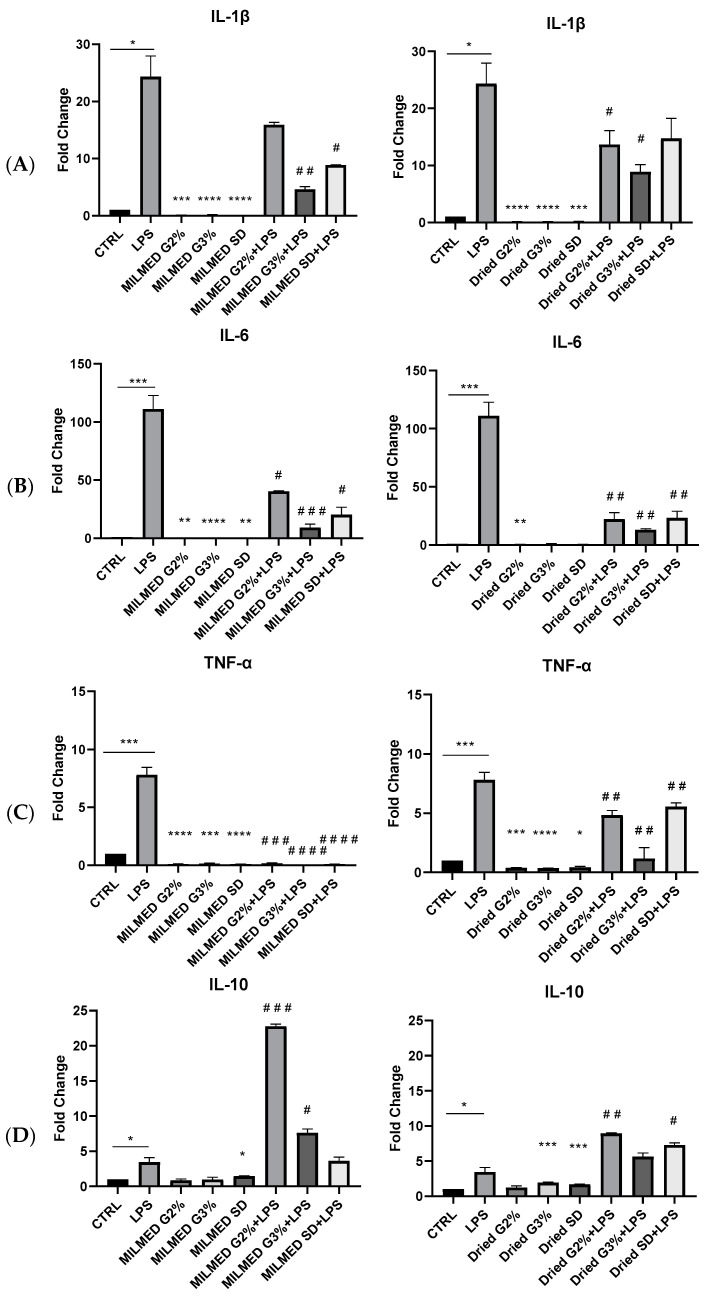
mRNA expression of (**A**) interleukin-1β (IL-1 β); (**B**) interleukin-6 (IL-6); (**C**) Tumor Necrosis Factor-α (TNF-a); and (**D**) interleukin-10 (IL-10); monitored by qRT-PCR and normalized to β-actin. The decrease of mRNA expression of pro-inflammatory cytokines is associated with increased mRNA expression of IL-10 in BV2 cells in both Milmed and Dried treatment. Data are shown as mean ± SD from three independent experiments performed in triplicate. Expression profiles were determined using the 2^−ΔΔCT^ method. Statistical analysis was evaluated by unpaired Student’s *t* test. * *p* < 0.05, ** *p* < 0.01. *** *p* < 0.001, **** *p* < 0.0001; ## *p* < 0.01; ### *p* < 0.001; #### *p* < 0.0001 * VS CTRL, # VS LPS.

**Table 1 biomedicines-10-03116-t001:** List of Primers.

** *Gene* **	**Forward Primer (5′–3′)**	**Reverse Primer (5′–3′)**	**Accession Numbers**
*mIL-1β*	GAAATGCCACCTTTTGACAGTG	TGGATGCTCTCATCAGGACAG	NM_008361.4
*mTNF-α*	CTGAACTTCGGGGTGATCGG	GGCTTGTCACTCGAATTTTGAGA	BC137720.1
*mIL-10*	GCCCTTTGCTATGGTGTCCTTTC	TCCCTGGTTTCTCTTCCCAAGAC	NM_010548.2
*mARG1*	ATGTGCCCTCTGTCTTTTAGGG	GGTCTCTCACGTCATACTCTGT	NM_007482.3
*miNOS*	GGCAGCCTGTGAGACCTTTG	GCATTGGAAGTGAAGCGTTTC	AF427516.1
*mIL-6*	CGGAGAGGAGACTTCACAGAGGA	TTTCCACGATTTCCCAGAGAACA	NM_001314054.1
*mACT-β*	GGCTGTATTCCCCTCCATCG	CCAGTTGGTAACAATGCCATGT	NM_007393.5

## Data Availability

Not applicable.

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
