# Peer review of "Milmed Yeast Alters the LPS-Induced M1 Microglia Cells to Form M2 Anti-Inflammatory Phenotype"

_biomedicines, 2022, doi:10.3390/biomedicines10123116_

Round 1

Reviewer 1 Report

The manuscript is adequately structured and presented. Specific comments:

1. Methods section. The authors need to add a subsection on statistical analyses performed and describe what tests were performed and what software was used for this. Also, since the authors used parametric analysis, this needs to be justified.

2. Results section. The authors need to move a sentence “Several chronic neurodegenerative diseases affecting the central nervous system have been associated with the polarization of microglia towards a proinflammatory phenotype; these are characterized generally by a change in the shape of the cells and by increased expression of iNOS, as well as an increased synthesis of proinflammatory cytokines” either to Introduction or discussion sections.

Otherwise, the manuscript is of interest to the audience of Biomedicines.  

Author Response

1.Methods section. The authors need to add a subsection on statistical analyses performed and describe what tests were performed and what software was used for this. Also, since the authors used parametric analysis, this needs to be justified.

We thank the reviewer for his suggestion We added a new paragraph dealing with the statistical analysis

Statistical Analysis

Data were expressed as the mean values ± standard deviations (SD) from at least three independent experiments. The statistical significance was determined by one-way ANOVA analysis coupled with a Bonferroni post-test or Unpaired T-test (GraphPad Prism 5.0 software, GraphPad Software Inc., La Jolla, CA, USA) and defined as * p < 0.05, ** p < 0.01, and *** p < 0.001. Ns is not significant.  Parametric statistics, one-way ANOVA and Bonferroni tests. were applied to the results since group size was held constant throughout and normal distributions were obtained.

  1. Results section. The authors need to move a sentence “Several chronic neurodegenerative diseases affecting the central nervous system have been associated with the polarization of microglia towards a proinflammatory phenotype; these are characterized generally by a change in the shape of the cells and by increased expression of iNOS, as well as an increased synthesis of proinflammatory cytokines” either to Introduction or discussion sections.

The sentence was moved to discussion, lines 293-296 of the revised version

Reviewer 2 Report

In this study, the authors showed that Milmed yeast, obtained from S. cerevisiae after exposure to electromagnetic millimeter wavelengths, induces a reversal of LPS- M1 polarized microglia towards an anti-inflammatory phenotype, as demonstrated morphologically by the recovery of resting phenotype by microglia, by the decrease in the mRNAs of IL-1 beta, IL-6, TNF-alpha and in the expression of iNOS.

This manuscript is interesting; nevertheless needs very substantial improvements and corrections before publishing may be possible.

General points:

Please add a list of abbreviations before References section to your manuscript.

Special points:

Please add the future perspectives sections to your manuscript and please describe the importance of your results for science and clinicians.

Introduction

Lines 28-30: please describe exactly all these sentences.

Lines 30-49: please add multiple references at the end of each these sentences.

Lines 70-80: please describe exactly all these studies.

Materials and Methods

Lines 83-85: please add multiple references at the end of this sentence.

Please add a product information for all products used in your study.

Please add according to which group or publication you did all methods used in this publication.  

Results

Lines 159-162: please add multiple references at the end of this sentence.

Figures

Please add to all Legends Figures 1-4 the abbreviation used in all Figures.

Discussion

 Lines 287-294: please describe all these studies very exactly.

Lines 294-296: please add multiple references at the end of this sentence.

Lines 294-302: please add multiple references at the end of each these sentences.

Lines 310-315: please add multiple references at the end of each these sentences.

Lines 333-336: please describe all these studies very exactly.

Author Response

Please add a list of abbreviations before References section to your manuscript.

A list of abbreviations was added before References section:

List of abbreviations

Alzheimer’s Disease (AD); Arginase-1 (Arg-1); Brain-Derived Neurotrophic Factor (BDNF); Central Nervous System (CNS); Dulbecco’s Modified Eagle’s Medium (DMEM); Extreme High Frequency (EHF); Fetal Bovine Serum (FBS); Immunoglobulins G (IgG); Inducible Nitric Oxide Synthase (iNOS); Interleukin-1β (IL-1β); Interleukin-6 (IL-6); Interleukin-10 (IL-10); Lipopolysaccharide (LPS); Nerve Growth Factor (NGF); Nuclear factor kappa-light-chain-enhancer of activated B cells (NF-κB); Parkinson’s disease (PD); Phalloidin-Tetramethylrhodamine B Isothiocyanate (Phalloidin- TRITC); .Real-Time PCR (qRT-PCR); Synthetic Defined Medium (SD); Threshold Cycle (TC); Tumor Necrosis Factor- α (TNF- α); Yeast Peptone (YP).

Please add the future perspectives sections to your manuscript and please describe the importance of your results for science and clinicians. 

(lines 379-384)

Future perspectives

Gut microbiota and their metabolites cooperate to the maintenance of homeostasis by interacting with innate and adaptive immune components, thereafter microbiome-targeted therapies such as probiotics, prebiotics and synbiotics hold great potentials to prevent and treat inflammatory diseases. The anti-inflammatory effects of the treated yeast preparation, Milmed, provide a plausible therapeutic application in the treatment of several areas of infection and/or allergic reactions, as recently indicated [17, 40]. In view of the anti-neurodegenerative observations of the treated yeast [21], further research upon the effects of the treated yeast upon growth factors and autophagy ought to be explored.

Introduction

Lines 28-30: please describe exactly all these sentences. 

According to reviewer’s suggestions we have elaborated the sentences adding details and new references

(Lines 30-38):

Neuroinflammation is now hypothesized to be the key mechanism of Alzheimer’s disease (AD), induced by the polarization of microglial cells towards a proinflammatory phenotype. Microglial cells play a central role in maintaining brain homeostasis and are involved in resolving inflammation from trauma or infectious microorganisms by means of phagocytosis and/or anti-inflammatory mediators. [ Grieco M, De Caris MG, Maggi E, Armeli F, Coccurello R, Bisogno T, D'Erme M, Maccarrone M, Mancini P, Businaro R. Fatty Acid Amide Hydrolase (FAAH) Inhibition Modulates Amyloid-Beta-Induced Microglia Polarization. Int J Mol Sci. 2021;22(14):7711. doi: 10.3390/ijms22147711].It has been recently shown that in the experimental models of AD treated with probiotics it was possible to observe a reduction in inflammatory processes as well as an increase in the level of antioxidant enzymes, and a decrease in beta-amyloid deposition as well as in tau hyperphosphorylation (Neta FI, de Souza FES, Batista AL, Pinheiro FI, Cobucci RN, Guzen FP. Effects of supplementation with probiotics in experimental models of Alzheimer's Disease: a systematic review of animal experiments. Curr Alzheimer Res. 2022 doi: 10.2174/1567205019666220318092003).

Lines 30-49: please add multiple references at the end of each these sentences.

(lines 41-49)

The importance of gut-brain axis has been recently uncovered and the relevance of a correct microbiota balance has been highlighted [Varela-Trinidad GU, Domínguez-Díaz C, Solórzano-Castanedo K, Íñiguez-Gutiérrez L, Hernández-Flores TJ, Fafutis-Morris M. Probiotics: Protecting Our Health from the Gut. Microorganisms. 2022;10:1428.]. An imbalance in the components of the microbiota can damage the intestinal barrier, affecting in a bidirectional way the CNS: indeed, microbiota alterations were shown to be directly associated with neuropsychiatric disorders, promoting the development of depression and dementia [4 Eltokhi A, Sommer IE. A Reciprocal Link Between Gut Microbiota, Inflammation and Depression: A Place for Probiotics? Front Neurosci. 2022;16:852506]. Many studies have suggested that imbalance of the gut microbial composition are associated to an increase in pro- inflammatory cytokines and oxidative stress, that underly chronic neuroinflammatory diseases [Li K, Ly K, Mehta S, Braithwaite A. Importance of crosstalk between the microbiota and the neuroimmune system for tissue homeostasis. Clin Transl Immunology. 2022;11(5):e1394]. In addition, perturbations to the gut microbiota were detected in neurodegenerative conditions such as Parkinson’s disease (PD) and Alzheimer’s disease (AD), a complex community of microrganisms influenced not only by host genetics, but also by diet and the environment [5–7].

In this connection, much interest was focused on probiotics, the beneficial living bacteria and yeast, that may rebalance the bacterial-fungal gut microbiome, and reduce inflammation [9–11]. Several studies reported the beneficial effect of Saccharomyces yeast supplementation, decreasing inflammation and oxidative stress: in 2021 Durmaz et al [12] reported that oral supplementation of probiotic S. boulardii before supraceliac aortic ischemia

Lines 70-80: please describe exactly all these studies.

(lines 77-92)

The purpose of present study was to assess the ability of Milmed to counteract the microglia polarization towards the M1 pro-inflammatory phenotype induced by LPS and to analyze Milmed propensity to counteract inflammation. Our previous results showed that Milmed Saccharomyces cerevisiae yeast [Milmed], treated with electromagnetic waves in the Extreme High Frequency (EHF) range of 30–300 GHz, does not produce cytotoxic effects neither upon microglia cells nor upon cells of neuronal origin. Moreover, when added to LPS-stimulated microglial cells, it promoted a remodeling of the cells that acquired the branched spindle-like morphology characteristic of quiescent unstimulated cells., losing some phylopodia.If added to human neuroblastoma cells, Milmed yeast promoted the expression of BDNF and NGF mRNAs [2].Furthermore, patients presenting allergic problems aged from 30 to 71 years-of-age, received Milmed supplementation for 4 to 13 weeks. The median treatment time was 9 weeks. After the treatment, After the test period, each patient rated the perceived improvement on allergy on a scale of 1 to 10 and the perceived improvement in general health on a scale of 1 to 10. there were both self-assessed allergy level significant improvements (8.38 ± 0.62) and self-assessed general health level improvements (7.38 ± 0.82) in comparison with the control group. The Milmed supplementation to allergy patients alleviated allergy symptoms related to the number of weeks the treated Milmed was ingested [17]. In the present study, the polarization of microglial cells was analysed measuring the cytokine and chemokine mRNAs expression, as well as the expression of M1 and M2 markers such as iNOS and Arginase-1 by real-time PCR and immunofluorescence assays.

Materials and Methods

Lines 83-85: please add multiple references at the end of this sentence.

(lines 94-96)

The treatment and preparation of S. cerevisiae with electromagnetic waves in the Extreme High Frequency (EHF) range of 30–300 GHz produces a treated yeast extract, given the name Milmed.

[Prokhorov, A. M. & E. M. Dianov (2001). "In Memory of Mikhail B. Golant". Technical Physics Volume 46, Number 8, 1068, doi:10.1134/1.1395134

Devyatkov ND, Golant MB, Betsky OV. Brief information for physicians about the physical characteristics of the processes occurring in the body under MM-wave therapy performed by installations "Jav-1", and the associated effects on the body of electromagnetic millimeter waves  Radio i Svyaz, Moscow, 1991.

Betsky, OV; Devyatkov, ND; Kislov, V. (2000). "Low Intensity Millimeter Waves in Medicine and Biology". Critical Reviews in Biomedical Engineering28 (1&2): 247–268.

Devyatkov ND, Gelvich EA et al. Apparatus and methods for microwave and RF heating for application in oncology. Soviet Physics Uspekhi (Advances in Physical Sciences). 1981, May, v 134, N 1, pp. 158—163.]

Please add a product information for all products used in your study.

Cell Culture and Treatment

The BV-2 murine microglial cell line, kindly provided by Dr. Mangino, Sapienza University of Rome (Italy), was cultured in Dulbecco’s modified Eagle’s medium (DMEM DMEM; Euroclone, Pero, MI, Italy), supplemented with 10% fetal bovine serum (FBS; Sigma-Aldrich, MO, USA) and 1% penicillin-streptomycin (Sigma-Aldrich, MO, USA), at 37â—¦C in a humidified incubator under 5% CO2, until they reached 90% confluence. Cells were seeded in 6 well plates (cell density of 106 cells/well) and incubated at 37°C with 5x102 of treated or untreated yeast grown in different media (YP 2% Glucose, or YP 3% Glycerol, SD 2% glucose) for 45 min, after 45 min BV-2 cells were treated with LPS 1ng/ml and compared to control.

 Cell Viability Assays

Trypan Blue exclusion assay

Cell viability was determined by Trypan-Blue exclusion assay. Trypan-blue (Euroclone, Pero, MI, Italy) exclusion assay is a simple and rapid method measuring cell viability that determines the number of viable cells and dead cells. It is based on the principle that live cells with an intact membrane are able to exclude the dye whereas dead cells without an intact membrane take up the dye. For the Trypan blue exclusion test, BV-2 were seeded onto 48-well plates at a density of 3 ×104/well. After treatments, cells were detached with 1× Tripsin-EDTA (Sigma-Aldrich, St. Louis, MO, USA), and 10 μL of cell suspension were mixed with 10 μL of Trypan Blue solution and cell counts were performed using a Burker chamber. Blue stained cells were considered nonviable.

Immunofluorescence microscopy

Cell cultures were stimulated with lipopolysaccharide (LPS), a prototypical microbial antigen in the presence or absence of Milmed in different dilutions. Briefly, BV2 cells grown in chamber-slides were incubated with Milmed grown in YP 2% Glucose, or YP 3% Glycerol, or SD 2% glucose or SD 3% glycerol, in the presence or in the absence of LPS (strain 0111:B4, Sigma.Aldrich,1 ng /ml) for 24 h. Cells were subsequently fixed in 4% paraformaldehyde in phosphate-buffered-saline (PBS) for 30 min at 25°C, followed by treatment with 0.1 M glycine in PBS for 20 min at 25°C and with 0.1% Triton X-100 in PBS for additional 5 min at 25°C to allow permeabilization. To analyze cytoskeletal actin reorganization, microglial cells were incubated with Phalloidin-Tetramethylrhodamine B isothiocyanate (Phalloidin- TRITC), (Sigma-Aldrich, St. Louis, MO, USA) 1:50 for 45 min at 25°C. For the detection of M1/M2 polarization markers, cells were incubated with rabbit polyclonal antibodies raised against iNOS (dil. 1:100—D6B65, Cell Signaling Technology, Danvers, MA, USA) or rabbit polyclonal IgG anti-Arg-1 (dil. 1:50—D4E3M, Cell Signaling Technology, Danvers, Ma, USA) and subsequently with anti-rabbit Alexa Fluor 488 secondary antibodies. Finally, the cells were marked with DAPI (Sigma-Aldrich, St. Louis, MO, USA) to highlight the nucleus. The fluorescence signal was analyzed using an Axio Observer inverted microscope, equipped with the ApoTome System (Carl Zeiss Inc., Ober Kochen, Germany). Cell area was quantified with ImageJ software.

The other sections of materials and methods already reported all product informations

Please add according to which group or publication you did all methods used in this publication.  

Results

Lines 159-162: please add multiple references at the end of this sentence.

(lines 293-296)

Several chronic neurodegenerative diseases affecting central nervous system have been associated with the polarization of microglia towards a proinflammatory phenotype; these are characterized generally by a change in the shape of the cells and by increased expression of iNOS, as well as an increased synthesis of proinflammatory cytokines [Cherry JD, Olschowka JA, O'Banion MK. Neuroinflammation and M2 microglia: the good, the bad, and the inflamed. J Neuroinflammation. 2014;11:98. doi: 10.1186/1742-2094-11-98;.Tang Y, Le W. Differential Roles of M1 and M2 Microglia in Neurodegenerative Diseases. Mol Neurobiol. 2016; 53:1181-1194. doi: 10.1007/s12035-014-9070-5; Dubbelaar ML, Kracht L, Eggen BJL, Boddeke EWGM. The Kaleidoscope of Microglial Phenotypes. Front Immunol. 2018;9:1753. doi: 10.3389/fimmu.2018.01753; Jurga AM, Paleczna M, Kuter KZ. Overview of General and Discriminating Markers of Differential Microglia Phenotypes. Front Cell Neurosci. 2020;14:198. doi: 10.3389/fncel.2020.00198; Kwon HS, Koh SH. Neuroinflammation in neurodegenerative disorders: the roles of microglia and astrocytes. Transl Neurodegener. 2020;9:42. doi: 10.1186/s40035-020-00221-2.]

Figures

Please add to all Legends Figures 1-4 the abbreviation used in all Figures.

Figure 1. A: BV2 cells area following Lipopolysaccharide (LPS) treatment (1ng/ml) for 24 hr in the presence or absence of untreated yeast grown in different cultural conditions. LPS induced a cell body enlargement which was even more evident in the presence of untreated yeast grown in G2%. B: BV2 cells area following LPS treatment (1ng/ml) for 24 hr in the presence or absence of Milmed: after Milmed addition cells showed a measure of area similar to that of control in Glucose 2%, Glycerol 3% and SD with LPS. Data are expressed as mean ± SD for each group (n=3). Statistical analysis was performed by the one-way analysis of variance (ANOVA) method coupled with the Bonferroni post-test. * VS CTRL, # VS LPS **p < 0.01; *** p < 0.001.

Figure 2. A: Immunofluorescence analysis of Inducible Nitric Oxide Synthase (iNOS) expression in BV-2 cells cultured in the presence of untreated yeast grown for 24 hr in different metabolic medium without Lipopolysaccharide (LPS )(on the left, first column) or after addition of LPS 1 ng/ml (second column) and B: Arginase-1 (ARG-1) expression in BV-2 cells cultured in the presence of Milmed grown in different metabolic medium without LPS (on the right, first column) or after addition of LPS 1 ng/ml (second column). Quantification of the median fluorescence intensity was performed by ImageJ software and data were expressed as histograms, normalized to the number of cells for field. 4′,6-diamidino-2-phenylindole(DAPI )was used to counterstain the nuclei. Data are expressed as mean ± SD for each group (n=3). Statistical analysis was performed by the one-way analysis of variance (ANOVA) coupled with the Bonferroni post-test. * VS CTRL, # VS LPS *: p < 0.05, **: p < 0.01; *** p < 0.001.

Figure 3. BV2 cells were pretreated with 5 x102Milmed/well grown in Yeast Extract–Peptone–Dextrose  YPD medium or regrown in YPD medium from a dried preparation for 45 min and after incubated with Lipopolysaccharide (LPS) 1 ng/ml. Inducible Nitric Oxide Synthase (iNOS) andArginase-1 (ARG-1) mRNAs expressions were evaluated by qRT-PCR at 4 hr and normalized to β-actin. Data are shown as mean ± SD from three independent experiments performed in triplicate. Statistical analysis was evaluated by unpaired Student t test. Expression profiles were determined using the 2−ΔΔCT method.  * p < 0.05, ** p < 0.01. *** p < 0.001, **** p < 0.0001. * VS CTRL, # VS LPS

Figure 4. mRNA expression of interleukin-1β (IL-1 ) (A), interleukin-6 (IL-6) (B), Tumor Necrosis Factor- α (TNF-a)  (C) and interleukin-10 (IL-10 )(D) monitored by qRT-PCR and normalized to β-actin. The decrease of mRNA expression of pro-inflammatory cytokines is associated with increased mRNA expression of IL-10 in BV2 cells in both Milmed and Dried treatment. Data are shown as mean ± SD from three independent experiments performed in triplicate. Expression profiles were determined using the 2−∆∆CT method. Statistical analysis was evaluated by unpaired Student t test. * p < 0.05, ** p < 0.01. *** p < 0.001, **** p < 0.0001. * VS CTRL, # VS Lipopolysaccharide (LPS)

Discussion

 Lines 287-294: please describe all these studies very exactly.

(LINES 288-296)

Microbiota is involved in regulating gastrointestinal (GI), immune, nervous system and metabolic homeostasis [Chidambaram SB, Essa MM, Rathipriya AG, Bishir M, Ray B, Mahalakshmi AM, Tousif AH, Sakharkar MK, Kashyap RS, Friedland RP, Monaghan TM. Gut dysbiosis, defective autophagy and altered immune responses in neurodegenerative diseases: Tales of a vicious cycle. Pharmacol Ther. 2022;231:107988. doi:10.1016/j.pharmthera.2021.107988.]. When disruptive alterations in the composition of the microbiota, termed dysbiosis, take place, the imbalance of gut microbiota associates with unhealth outcome, affecting not only the health and integrity of the gastrointestinal tract but promoting also the onset of neuropsychiatric/neurologic diseases [4, 23]. Dysbiosis of gut microbiota can induce increased intestinal permeability, immune activation leading to systemic inflammation, which in turn may impair the blood-brain barrier and promote neuroinflammation, neural injury, and ultimately neurodegeneration. [ Goyal D, Ali SA, Singh RK. Emerging role of gut microbiota in modulation of neuroinflammation and neurodegeneration with emphasis on Alzheimer's disease. Prog Neuropsychopharmacol Biol Psychiatry. 2021;106:110112. doi: 10.1016/j.pnpbp.2020.110112; Kowalski K, Mulak A. Brain-Gut-Microbiota Axis in Alzheimer's Disease. J Neurogastroenterol Motil. 2019;25:48-60. doi: 10.5056/jnm18087.] The interconnections of gut microbiota perturbation with CNS disorders have been well-documented in recent times, as well as the real and /or putative interactions with the immune system [24, 25; Chen Z, Maqbool J, Sajid F, Hussain G, Sun T. Human gut microbiota and its association with pathogenesis and treatments of neurodegenerative diseases. Microb Pathog. 2021;150:104675. doi: 10.1016/j.micpath.2020.104675. E.].

Lines 294-302: please add multiple references at the end of each these sentences.

Lines 298-300

Furthermore, the direct impact of gut microbiota on the functions of microglial cells has been shown [24, 26]. Activated microglial cells play a central role in the development of many CNS diseases, including depression, AD, and PD. [ Kwon HS, Koh SH. Neuroinflammation in neurodegenerative disorders: the roles of microglia and astrocytes. Transl Neurodegener. 2020;9:42. doi: 10.1186/s40035-020-00221-2; Rahimian R, Belliveau C, Chen R, Mechawar N. Microglial Inflammatory-Metabolic Pathways and Their Potential Therapeutic Implication in Major Depressive Disorder. Front Psychiatry. 2022;13:871997. doi: 10.3389/fpsyt.2022.871997; Princiotta Cariddi L, Mauri M, Cosentino M, Versino M, Marino F. Alzheimer's Disease: From Immune Homeostasis to Neuroinflammatory Condition. Int J Mol Sci. 2022;23:13008. doi: 10.3390/ijms232113008; Choi I, Heaton GR, Lee YK, Yue Z. Regulation of α-synuclein homeostasis and inflammasome activation by microglial autophagy. Sci Adv. 2022;8(43):eabn1298. doi: 10.1126/sciadv.abn1298; Huang B, Zhenxin Y, Chen S, Tan Z, Zong Z, Zhang H, Xiong X. The Innate and Adaptive Immune Cells in Alzheimer's and Parkinson's Diseases. Oxid Med Cell Longev. 2022;2022:1315248. doi: 10.1155/2022/1315248. Araújo B, Caridade-Silva R, Soares-Guedes C, Martins-Macedo J, Gomes ED, Monteiro S, Teixeira FG. Neuroinflammation and Parkinson's Disease-From Neurodegeneration to Therapeutic Opportunities. Cells. 2022;11:2908. doi: 10.3390/cells11182908]. Their release of cytokines and chemokines target neuronal cells leading to cellular dysfunctionality and eventual cell death. [Thakur S, Dhapola R, Sarma P, Medhi B, Reddy DH. Neuroinflammation in Alzheimer's Disease: Current Progress in Molecular Signaling and Therapeutics. Inflammation. 2022 doi: 10.1007/s10753-022-01721-1]. Probiotic agents augment an improvement of depression symptoms and stress-related diseases modulated by stress hormones and cytokines release [27] in conjunction with microbiota influencing the ability of neurons, brain tissue and immune cells to regulate inflammation [28]. Clinical and experimental data have indicated that the probiotics induce positive impacts upon the central nervous system. [ Chen Y, Peng F, Xing Z, Chen J, Peng C, Li D. Beneficial effects of natural flavonoids on neuroinflammation. Front Immunol. 2022;13:1006434. doi: 10.3389/fimmu.2022.1006434; Marć MA, JastrzÄ…b R, Mytych J. Does the Gut Microbial Metabolome Really Matter? The Connection between GUT Metabolome and Neurological Disorders. Nutrients. 2022;14:3967. doi: 10.3390/nu14193967].

Lines 310-315: please add multiple references at the end of each these sentences.

Lines 315-316

The present results imply that the Milmed preparation of S. cerevisiae, treated with electromagnetic wavelengths in the Extreme High Frequency – millimeter range [2; Betsky, OV; Devyatkov, ND; Kislov, V. (2000). "Low Intensity Millimeter Waves in Medicine and Biology". Critical Reviews in Biomedical Engineering28 (1&2): 247–268; Archer T, Fredriksson A. The yeast product Milmed enhances the effect of physical exercise on motor performance and dopamine neurochemistry recovery in MPTP-lesioned mice. Neurotox Res. 2013;24:393-406. doi: 10.1007/s12640-013-9405-4; Archer T, Garcia D, Fredriksson A. Restoration of MPTP-induced deficits by exercise and Milmed(®) co-treatment. PeerJ. 2014;2:e531. doi: 10.7717/peerj.531], present a probiotic agent that drives the innate immune cells’ polarization disposition. According to this scenario, the decrease in cytokine release after LPS inflammatory stimulus and the polarization of microglial cells towards an anti-inflammatory phenotype, as confirmed by the induction of IL-10 and arginase-1, following Milmed addition leads to the assertion that Milmed supplementation offers a promising treatment for pathologies arising from neuro-inflammation, such as depression, Parkinsonism and AD.

Lines 333-336: please describe all these studies very exactly

Lines 334-345

Moreover, numerous recent reports have highlighted the possibility of using compounds of vegetable origin, known as nutraceuticals, or particular diets to treat inflammation and to drive macrophage polarization towards an anti-inflammatory phenotype [33–36]. The evidence of a microbiota–gut–brain axis suggests that modulation of the gut microbiota may be a tractable strategy for developing novel therapeutics for complex CNS disorders, and dietary habits play a crucial role in the selection of the bacterial community in the human gastrointestinal tract [33] Plant-derived molecules such as resveratrol, was shown to target macrophage-related inflammation associated with neurodegenerative diseases: it was shown to  suppress the LPS-stimulated microglia release of proinflammatory mediators such as NO, TNF-α, iNOS, IL-1β, and IL-6, associated to M1 proinflammatory phenotype, and switch the cells to the expression of Arg1, CD163, and IL-10, markers associated to the M2 anti-inflammatory phenotype [Wang L, Zhao H, Wang L, Tao Y, Du G, Guan W, Liu J, Brennan C, Ho CT, Li S. Effects of Selected Resveratrol Analogues on Activation and Polarization of Lipopolysaccharide-Stimulated BV-2 Microglial Cells. J Agric Food Chem. 2020;68:3750-3757. doi: 10.1021/acs.jafc.0c00498]. Moreover, resveratrol was shown to counteract the 7-oxo-cholesterol-triggered proinflammatory signaling in macrophages. [Buttari B, Profumo E, Segoni L, D'Arcangelo D, Rossi S, Facchiano F, Saso L, Businaro R, Iuliano L, Riganò R. Resveratrol counteracts inflammation in human M1 and M2 macrophages upon challenge with 7-oxo-cholesterol: potential therapeutic implications in atherosclerosis. Oxid Med Cell Longev. 2014;2014:257543. doi: 10.1155/2014/257543]. Several nutritional protocols, including Mediterranean diet and ketogenic diets, were able to counteract neuroinflammation in cellular experimental models as well as in clinical trials [Román GC, Jackson RE, Gadhia R, Román AN, Reis J. Mediterranean diet: The role of long-chain ω-3 fatty acids in fish; polyphenols in fruits, vegetables, cereals, coffee, tea, cacao and wine; probiotics and vitamins in prevention of stroke, age-related cognitive decline, and Alzheimer disease. Rev Neurol (Paris). 2019;175:724-741. doi: 10.1016/j.neurol.2019.08.005; Businaro, R.; Vauzour, D.; Sarris, J.; Münch, G.; Gyengesi, E.; Brogelli, L.; Zuzarte, P. Therapeutic Opportunities for Food Supplements in Neurodegenerative Disease and Depression. Front. Nutr. 2021, 8, 669846, doi:10.3389/fnut.2021.669846; De Caris, M.G.; Grieco, M.; Maggi, E.; Francioso, A.; Armeli, F.; Mosca, L.; Pinto, A.; D’Erme, M.; Mancini, P.; Businaro, R. Blueberry Counteracts BV-2 Microglia Morphological and Functional Switch after LPS Challenge. Nutrients 2020, 12, 1830, doi:10.3390/nu12061830; Angeloni, C.; Businaro, R.; Vauzour, D. The Role of Diet in Preventing and Reducing Cognitive Decline. Current Opinion in Psychiatry 2020, 33, 432–438, doi:10.1097/YCO.0000000000000605; Businaro, R.; Corsi, M.; Asprino, R.; Di Lorenzo, C.; Laskin, D.; Corbo, R.M.; Ricci, S.; Pinto, A. Modulation of Inflammation as a Way of Delaying Alzheimer’s Disease Progression: The Diet’s Role. CAR 2018, 15, 363–380, doi:10.2174/1567205014666170829100100; Pinto A, Bonucci A, Maggi E, Corsi M, Businaro R. Anti-Oxidant and Anti-Inflammatory Activity of Ketogenic Diet: New Perspectives for Neuroprotection in Alzheimer's Disease. Antioxidants (Basel). 2018;7:63. doi: 10.3390/antiox7050063; Versele R, Corsi M, Fuso A, Sevin E, Businaro R, Gosselet F, Fenart L, Candela P. Ketone Bodies Promote Amyloid-β1-40 Clearance in a Human in Vitro Blood-Brain Barrier Model. Int J Mol Sci. 2020;21:934. doi: 10.3390/ijms21030934].

Reviewer 3 Report

The manuscript entitled "Milmed yeast alters the LPS-induced M1 microglia cells to form 2 M2 anti-inflammatory phenotype" is a search for a causative factor in the development of e.g. neurodegenerative diseases such as AD and PD and is part of the current research on this pathology.

Strengths

innovative research direction

Weaknesses

no research conclusions.

no probable use of research, e.g. predictive and therapeutic purposes

My comments:

Introduction:

Add the share of potential microbiota in AD and PD, e.g. Piekut et al., 2022

Improve the purpose of the work, no expected research results. Only test methods are given.

Describe the relationship between the examined individual biochemical parameters.

Result:

Mark the analyzed cell changes on the micrographs.

Discussion:

…diseases, including 295

depression, AD, and PD. Their release of cytokines and chemokines target neuronal cells 296

leading to cellular dysfunctionality and eventual cell death. Probiotic agents augment an…..

in what mechanism, supplement.

…Clinical and ex- 300

perimental data have indicated that the probiotics induce positive impacts upon the cen- 301

tral nervous system….

in what mechanism, supplement.

Conclusion:

Improve the conclusion, e.g. how Milmed factors improve the level of individual biochemical parameters.

Correct linguistic error.

Author Response

The manuscript entitled "Milmed yeast alters the LPS-induced M1 microglia cells to form 2 M2 anti-inflammatory phenotype" is a search for a causative factor in the development of e.g. neurodegenerative diseases such as AD and PD and is part of the current research on this pathology.

Strengths

innovative research direction

Weaknesses

no research conclusions.

no probable use of research, e.g. predictive and therapeutic purposes

According to reviewer suggestions we added a new paragraph dealing with conclusions and perspectives:

Lines 378-384

Future perspectives

Gut microbiota and their metabolites cooperate to the maintenance of homeostasis by interacting with innate and adaptive immune components, thereafter microbiome-targeted therapies such as probiotics, prebiotics and synbiotics hold great potentials to prevent and treat inflammatory diseases. The anti-inflammatory effects of the treated yeast preparation, Milmed, provide a plausible therapeutic application in the treatment of several areas of infection and/or allergic reactions, as recently indicated [17, 40]. In view of the anti-neurodegenerative observations of the treated yeast [21], further research upon the effects of the treated yeast upon growth factors and autophagy ought to be explored.

My comments:

Introduction:

Add the share of potential microbiota in AD and PD, e.g. Piekut et al., 2022

According to the reviewer’s suggestions we quoted the paper by  Piekut et al, 2022 in the introduction (lines 54-57): 

In addition it has recently been hypothesized that microorganisms may contribute to the development of Alzheimer's disease either by producing amyloid-like molecules or enhancing the production of endogenous Abeta, or by increasing systemic inflammation that targeting glial cells contributes to the neuronal damage observed during Alzheimer's disease (Piekut T, Hurła M, Banaszek N, Szejn P, Dorszewska J, Kozubski W, Prendecki M. Infectious agents and Alzheimer's disease. J Integr Neurosci. 2022;21:73. doi: 10.31083/j.jin2102073)

Improve the purpose of the work, no expected research results. Only test methods are given

According to reviewer’s suggestions a sentence focused on the purpose of the study was added (lines 77-78):

The purpose of present study was to assess the ability of Milmed to counteract the microglia polarization towards the M1 pro-inflammatory phenotype induced by LPS and to analyze Milmed propensity to counteract inflammation.

Describe the relationship between the examined individual biochemical parameters.

At this aim M1 and M2 markers, that are IL-1b, Il-6, TNF-a, iNOS for M1 phenotype, and IL-10, arginase-1 for M2 phenotype, were analyzed by several methods. As expected, a decrease of inflammatory markers (M1) was paralleled by an increase of anti-inflammatory markers (M2).

Result:by Piekut et al

Mark the analyzed cell changes on the micrographs.

The micrographs of figure 2 show the expression and localization of iNOS and ARG-1 proteins, as detected by immunofluorescence positivity. Since this type of fluorescent signal does not allow to clearly evaluate the morphological variations that the cells undergo after LPS treatment, we cannot mark the cellular variations on these micrographs. Moreover, because the morphological changes have already been described in the article quoted in the references [2], in the present study we have added only the quantitative data relating to the cell size differences after LPS treatments, in the presence or in the absence of Milmed, When LPS-treated BV-2 cells were supplemented with MILMED, there was a reduction in cell area that revert to control phenotype, suggesting yeast promotes the recovery of a resting phenotype.  

Discussion:

…diseases, including 295

depression, AD, and PD. Their release of cytokines and chemokines target neuronal cells 296

leading to cellular dysfunctionality and eventual cell death. Probiotic agents augment an…..

in what mechanism, supplement.

We wish to thank the referee for his comment: we added the following paragraph (lines 34-38) :

It has been recently shown that in the experimental models of AD treated with probiotics it was possible to observe a reduction in inflammatory processes as well as an increase in the level of antioxidant enzymes, and a decrease in beta-amyloid deposition as well as in tau hyperphosphorylation (Neta FI, de Souza FES, Batista AL, Pinheiro FI, Cobucci RN, Guzen FP. Effects of supplementation with probiotics in experimental models of Alzheimer's Disease: a systematic review of animal experiments. Curr Alzheimer Res. 2022 doi: 10.2174/1567205019666220318092003).

…Clinical and ex- 300

Experimental data have indicated that the probiotics induce positive impacts upon the cen- 301

tral nervous system….

in what mechanism, supplement.

According to the reviewer’s suggestion we included the following new paragraph (lines 305-310):

Probiotics were shown to reduce CNS inflammatory processes and activation of microglia, decreasing neuroinflammation, probably through the production of the brain-derived neurotrophic factor (BDNF) and the improved synaptic plasticity, related to on AD patients the increased concentration of  postsynaptic density protein 95 (Zhu G, Zhao J, Zhang H, Chen W, Wang G. Administration of Bifidobacterium breve Improves the Brain Function of Aβ1-42-Treated Mice via the Modulation of the Gut Microbiome. Nutrients. 2021;13:1602. doi: 10.3390/nu13051602). A  randomized, double-blind, and controlled clinical trial on AD patients showed that a 12-week probiotic treatment induced a significant improvement in the MMSE score  and a significant decrease in plasma malondialdehyde in the probiotic-treated group  (Akbari E, Asemi Z, Daneshvar Kakhaki R, Bahmani F, Kouchaki E, Tamtaji OR, Hamidi GA, Salami M. Effect of Probiotic Supplementation on Cognitive Function and Metabolic Status in Alzheimer's Disease: A Randomized, Double-Blind and Controlled Trial. Front Aging Neurosci. 2016;8:256. doi: 10.3389/fnagi.2016.00256.)

Conclusion:

Improve the conclusion, e.g. how Milmed factors improve the level of individual biochemical parameters.

We added new details to the conclusion (lines 368-384):

Conclusions

The treated yeast, Milmed, has been shown in several studies to exert anti-inflammatory actions against the proinflammatory action of LPS, as indicated above.These present findings indicated that Milmed induced both (i) an anti-pro-inflammatory effect, i.e. reduction of iNOS that LPS exposure had caused, and (ii) a direct anti-inflammatory action arising from enhanced ARG-1 expression. Further anti-inflammatory effects were induced through the antagonism of the pro-inflammatory effects of the pro-inflammatory cytokines, IL-1β, IL-6 and TFN-α, together with the enhanced expression of the anti-inflammatory cytokine, IL-10.

Limitations: The status of Milmed as a probiotic agent requires characterization with regard to distribution and fate in vivo.

Future perspectives

Gut microbiota and their metabolites cooperate to the maintenance of homeostasis by interacting with innate and adaptive immune components, thereafter microbiome-targeted therapies such as probiotics, prebiotics and synbiotics hold great potentials to prevent and treat inflammatory diseases. The anti-inflammatory effects of the treated yeast preparation, Milmed, provide a plausible therapeutic application in the treatment of several areas of infection and/or allergic reactions, as recently indicated [17, 40]. In view of the anti-neurodegenerative observations of the treated yeast [21], further research upon the effects of the treated yeast upon growth factors and autophagy ought to be explored

Correct linguistic error. 

The English mother tongue Authors have carefully revised the manuscript

Round 2

Reviewer 2 Report

Thank you for your corrections.